# Spinopelvic Alignment and Its Use in Total Hip Replacement Preoperative Planning—Decision Making Guide and Literature Review

**DOI:** 10.3390/jcm10163528

**Published:** 2021-08-11

**Authors:** Piotr Stępiński, Artur Stolarczyk, Bartosz Maciąg, Krzysztof Modzelewski, Jakub Szymczak, Weronika Michalczyk, Julia Zdun, Szymon Grzegorzewski

**Affiliations:** 1Department of Orthopedics and Rehabilitation, Medical University of Warsaw, 02-091 Warsaw, Poland; artur.stolarczyk@wum.edu.pl (A.S.); bmaciag4@wum.edu.com (B.M.); kmodzelewski@wum.edu.pl (K.M.); jakubszymczak92@gmail.com (J.S.); 2Medical University of Warsaw, 02-091 Warsaw, Poland; michalczyk.wa@gmail.com (W.M.); julkazdunka@gmail.com (J.Z.); s.grzegorzewski1999@gmail.com (S.G.)

**Keywords:** spinopelvic, alignment, total, hip, replacement, alloplasty, preoperative, planning

## Abstract

Worldwide tendencies to perform large numbers of total hip arthroplasties in the treatment of osteoarthritis are observable over a long period of time. Every year, there is an observable increase in the number of these procedures performed. The outcomes are good but not ideal, especially in groups of patients with spine problems. In recent years, a growing interest in this field may be observed, since spinopelvic alignment seems to have a significant impact on total hip replacement (THR) results. The aim of this study is to describe relations between spine and pelvic alignment and provide practical information about its impact on total hip replacement. The authors performed a literature review based on PubMed, Embase, and Medline and provide practical guidelines based on them and their own experience.

## 1. Introduction

Osteoarthritis (OA) affects one in three people over the age of 65, and it is more common among women than men. This multifactorial disease leads to structural changes of the joint, and it is connected to chronic conditions. OA is characterized by pain, stiffness, and decreased range of motion (ROM). These factors lead to poor quality of life—insomnia, depression, lack of confidence, and limitations in daily activities, work, or hobbies. OA causes very serious problems for patients and significant social and economic costs [1].

Total hip replacement (THR) was a revolutionary method used for the treatment of an end-stage osteoarthritis in the hip. The aim of this operation is to increase the patient’s range of motion and activity level, alleviate pain, reduce limitations in everyday life, and, ultimately, improve the patients’ standard of living [2]. Although the first steps in modern THR date back to the 1940s [3], this technique is constantly enhanced. It should also be pointed out that indications of THR have changed throughout the years. In the past, this procedure was reserved for infirm, ailing people having major difficulties walking. Nowadays, the range of indications is much wider. Contemporary technologies are able to deliver highly advanced implants to meet even the most demanding requirements and assure patients’ fully functioning life, full of challenging activities [4]. Our knowledge of total hip replacement, according to the records, is enriched with the classification of architectural hip deformities [5], perioperative care [6], and the use of alternative types of articulations, e.g., dual mobility components [7]. Currently, we can also feature many different bearing types used for THR [8], which improves patient outcomes after an operation.

Despite great advancements and fantastic results of the majority of operations, approximately 10% of patients are still not satisfied with the effect of THR [9,10]. Searching for the reasons of discontent, the following problems seem to play a vital role: insufficient restoration of ROM, perceptible distinction between the length of lower extremities [5], dislocation of prosthesis elements, and need for revision surgery. The key to achieve satisfaction of patients and perform successful THR with positive results is a traditional or digital preoperative strategy, which has been emphasized by many authors throughout the years [5,11,12,13,14].

During the last few years, there has been a growing interest in parameters called “spinopelvic alignment”. When it comes to preoperative planning before total hip replacement, hip–spine relations seem to play a big role and have been underestimated during recent years. Interest in that relation is growing as it becomes more clear that it has major clinical consequences [15], especially in the risk of dislocations [16]. This study is designed to provide practical advice on preoperative planning for total hip replacement.

## 2. Materials and Methods

A review of the literature was performed. A search for all articles connected with the topic, with the time frame set to 1900–2021, was performed. Keywords relating to spinopelvic alignment and total hip replacement were searched using the following online databases: Embase, Medline, and PubMed. Search filters included English language studies, research on humans, articles in press, and available abstracts. Papers were included in this review based on their titles; then, their abstract; and, finally, the full paper was assessed. The exclusion criteria were: only the abstract or title were available; the study was in any language other than English; the article was ahead of print; and the study concerned animals. On the next level of preparation, studies were excluded if they did not include information about spinopelvic alignment in total hip replacement or information about it was not relevant. The full text of the articles that met these criteria were obtained; then, a manual search was performed. Finally, papers not up to date with historical findings were excluded, as well as duplicates. A search and risk of bias assessment was performed by a single researcher. Data was extracted from the articles by one author and rechecked by the same author. Analysis and synthesis of studies were performed by one author.

## 3. What May Influence the Spinopelvic Alignment?

The hip and spine coexist in a biomechanical chain, and require special coordination between them. The lumbosacral joint connects the pelvis with the spine. On each side of the body, the hip joint and sacroiliac joint form spine–pelvic–hip connections, which are crucial in pelvic motion and maintaining appropriate balance during bipedal locomotion. Every ongoing disease process associated with joints mentioned before restricts mobility, decreases stability, and makes activities of daily life difficult.

With age, and due to other conditions, such as osteoarthritis, osteoporosis, or fractures, spinal curvatures evolve, mostly causing an incorrect spinal position and imbalance. The sagittal imbalance has a connection with disability and pain and occurs as a result of decreased lumbar lordosis, increased thoracic kyphosis, contractures in hips or knees, and changes in pelvic parameters mentioned above. Human organisms adjust to the environment and develop compensatory mechanisms to prevent consequences of disbalance [17]. Over the course of a lifetime, compensatory mechanisms are exhausted, which causes pelvic retroversion—the pelvis becomes more horizontal, thinner, and wider. Cervical lordosis, lumbar lordosis, and thoracic kyphosis may become shallower or deeper. Most frequently, lumbar hypolordosis, resulting in hip extension, knee flexion, and ankle flexion [18,19], thoracic hyperkyphosis, and anterior spinal instability occur. This results in inevitable pathologies involving the axial skeleton, hip joints, knee joints, and ankles.

The interaction of the spine with the lower limbs occurs through the pelvis. The mobility of the pelvis acts as a “hinge” between the spine and the hip—it allows one to move upright on the lower limbs [20].

Medical procedures performed on patients should also be taken into account. It was demonstrated that spinal fusion before THR might increase the risk of dislocation and impingement by increasing posterior pelvic tilt [1]. Nevertheless, the more segments are involved, the higher the limitations and the more dysfunctional the hip–spine biomechanics are [1,2].

## 4. Problems Associated with Improper Spinopelvic Mobility and THR

Dislocation of a hip prosthesis is a common complication occurring after the THR. The rate varies from 0.2 to 10% per year [21]. Even in 1980, there were reports of the impact of neuromuscular and cognitive disorders or excessive intake of alcohol beverages on the prevalence of single or recurrent dislocation [22]. Dislocations have a range of other risk factors, such as older age [23], gender, comorbidities such as rheumatoid arthritis (RA) [24], or surgical approach [25]. Another very important role in dislocations after THR is cup and stem position. The reports indicate a correlation between prevalence of posterior dislocation and low cup anteversion [23]. The size of the femoral head articulation is also instrumental in decreasing the risk of dislocation. Larger, 36 mm femoral heads, compared with smaller, 28 mm articulations, lower the incidence of displacement during the first year after primary THR [26].

The “safe zone” (anteversion 15° ± 10°, inclination 40° ± 10° of acetabular cup), defined by Lewinnek, was designed to decrease the risk of dislocation after primary THR [16]. However, dislocations still occur [27]. One of the main reasons for that is probably the spine dynamics. Patients with a sagittal spinal deformity (SSD) are not protected by the “safe zone” [28]. SSD means abnormal kyphosis or lordosis, which can result in abnormalities within the pelvis [29].

Since spine dynamics are not the only risk factor, surgeons must be aware of the other ones. Unfortunately, almost all of them cannot be fixed by preoperative planning and special positioning of implants. That is why the surgeon should pay special attention to spine dynamics—one of the most important factors, and one of very few amenable to change by the orthopedic surgeon.

## 5. Anatomy and Imaging

Before starting the operation, proper planning should be done. Normally, the whole process is done based on antero-posterior pelvic X-ray. In case of any suspicion of abnormalities with spinopelvic alignment, special lateral views could also be useful, as they allow one to perform measurements of more sophisticated parameters of pelvic alignment. This should visualize a part of the body from L1 to the proximal femur, including the pelvis. Example of such X-rays are seen in Figure 1.

### Radiographic Measurements

Sacral Slope (SS)—to measure this angle, one needs to draw the straight line of the S1 superior endplate and a leveled line at a right angle to the gravitational force direction (horizontal reference line) [30]. The normal value ranges between 32 and 49° [31].Pelvic Tilt (PT)—an angle between the reference vertical line and the line joining the middle of S1 upper endplate and the center of the femoral head. The normal value ranges from 7 to 19° [31].Pelvic Incidence (PI)— the angle between the line that is formed by connecting the upper endplate of S1 (at its midpoint) to the femoral head axis. The normal value ranges from 38 to 56° [31].Pelvic Femoral Angle (PFA)—the position of the femur in relation to the pelvis. It is the angle centered at the femoral head, between the mid sacral base and down femoral shaft. The normal value ranges from 1 to 17° [31].Lumbar Lordosis (LL)—the segmental angle of spinal segment in lordosis, measured between the line on the upper endplate of L1 and the line on the upper endplate of S1(L1 -L5). The normal value ranges from 40 to 58° [31].Femoral Inclination (FI)—the angle between a vertical reference line and the axis of the femur. The normal value ranges from 0 to 8° [31].Sacro Femoral Angle (SFA)—the angle between the line of the upper endplate of S1 and the axis of the femur. The normal value ranges from 43 to 61° [31]Spino Sacral Angle (SSA)—the angle between the line of the upper endplate of S1 and a reference vertical line. The normal value ranges from 119 to 133° [31].

All angles mentioned above refer to an X-ray in standing position, which is the most common way to take radiographs for preoperative planning. It is also advisable to take the radiographs in sitting upright position to view the changing relations between the angles. It should be taken into account that the position of the torso during sitting may influence spinopelvic alignment and have an impact on the hip joint [32].

## 6. Approach to Preoperative Planning—Spinopelvic Mobility

When it comes to planning a total hip replacement, surgeons must be aware that the position of the acetabulum changes and depends on the position of the patient. In many cases, surgeons measure the pelvic tilt just before acetabular reaming while the patient is lying in supine position. It should be stressed that the patient is not going to stay in this position the whole time, and placement of the acetabulum should be fitted not only to this position (which slightly differs from standing position), but also to a sitting one.

While changing position from standing to sitting, numerous changes are observed between spine and pelvis (Table 1). While lumbar lordosis decreases, the sacrum moves backward. This leads to an increase in acetabular anteversion [33]. This shows that only part of the movement is performed by the hip joint. Range of movement depends on spinopelvic mobility. In the literature, there are described three types, each of them with a different impact:

Not all parameters are equally important in preoperative planning. Essential parts of each of them are connected with one another, giving the following equation:Pelvic Incidence (PI) = Sacral Slope (SS) + Pelvic Tilt (PT)

There is no need to measure every angle mentioned above. When it comes to other, more sophisticated ones, they describe more complicated relations between pelvis and spine and are not essential parts of the preoperative planning, although the knowledge may provide a broader view. Since pelvic tilt plays a vital role in the concept of spinopelvic alignment, it is considered the most important parameter [34]. Moreover, other parameters are in close relation to those essential ones from the quotation above, e.g., a decrease of 1° ∆ SS results in a 0.9° increase in pelvic femoral angle (PFA) [35]. All those parameters describe a pelvic position which is an important information in preoperative planning (Table 2).

## 7. Decision Making Proposition

During the last two years, our joint replacement team, which included three high volume surgeons (with over 2000 THR done), performed over 700 joint replacement surgeries, including over 250 total hip replacements. So far, we have noticed only one dislocation of traumatic etiology in a neurologically impaired patient. The recommendations described below (Table 3) include practical tips based on their knowledge and experience, which are supported by the newest literature.

It is still unknown what the optimal angles of anteversion or inclination are in particular cases of alterations in spinopelvic alignment. The optimal angles, earlier described as the “safe zone”[16], also change and no longer give optimal results [15]. To get better results and “safety”, the total hip replacement surgeon, based on recommendations in the table above and their own clinical experience, should decide on the optimal positioning of acetabular component.

## 8. Discussion

To achieve satisfactory results after THR, a good understanding of the spinopelvic motion, acetabulum location, and risk factors of impingement are needed. So far, many important investigations have been performed in this field. One of the most important was to find a relationship between pelvic tilt (PT) and acetabulum. The growth in anteversion of the acetabular cup for each degree posterior PT equals approximately 0.7°, which clearly shows the impact of PT on the position of the cup [36]. This parameter seems to be crucial during preoperative planning and was used in the algorithm. Another equally important parameter is cup inclination, which increases about 0.3 degrees with 1 degree of increase in pelvic tilt, although this relationship seems to be nonlinear and more complicated, as inclination is more connected with anteversion, and the change does not seem to be linear [37].

Apart from that, it is important to realize that dislocation may not only be correlated with spinopelvic alignment, but also with other factors listed previously [23,24,25]. The overall risk of dislocation is probably connected with all of them.

Another issue that is worth attention is spinal fusion in relation to THR. The latter should be a priority. Afterwards, if there are indications, spinal fusion can be carried out. There is one exception in which the order should be reversed. The spinal fusion may be done earlier than THR when the patient’s pelvic retroversion is increased [1,2]. Another approach may cause risk to THR stability [38]. As noted in the literature, the dislocation rate in patients with lumbar spine fusion (LSF) after THR is 1.7%, in patients with THR without spine pathology is 2.3%, in patients with THR with spine pathology at 3.3%, and in patients with THR after LSF is 4.6% [39]. Other works give more information regarding how not only the timing of the operations but also the range of lumbar fusions can affect the dislocation rate. It is estimated that the dislocation rate for THR patients without prior spinal fusion was 2.4%, 4.3% for patients with one to two levels fused, and 7.5% for patients with three to five levels fused [40].

New technologies of, and approaches to, hip alloplasty allow surgeons to operate on patients even with a high degree of stiffness between spine and pelvis. If the patient’s ante-inclination values change less than 5 degrees between sitting and standing position (acetabulum does not accommodate during spinopelvic motion), there is a higher risk of dislocation. In this case, the surgeon has to use a dual mobility articulation cup, making THR. Dual mobility cup (DMC) comprises a small head connected to a polyethylene liner. The aim of DMC is to enhance ROM and better stabilization [41]. It provides a lesser risk of dislocation in cases more prone to that complication—especially in case of neurological diseases, accompanied by increased or decreased muscle tonicity, e.g., multiple sclerosis, or bone defects [41].

The literature review performed in this study excluded data that was not in English, or without abstract or title. That is one of the main limitations. Another limitation is a quality of the study. Some studies enrolled in this review were performed on relatively small groups of patients; the results of the same studies performed on bigger groups may slightly differ.

To our best knowledge, there is no data concerning the impact of total knee replacement or foot and ankle problems on spinopelvic alignment. Since all parts of the lower limb may affect the result of THR, further research is needed in this field [18,19].

## 9. Conclusions

Spinopelvic alignment seems to play a vital role in understanding hip joint biomechanics and the impact of residual changes in the spine on hip mobility. This knowledge is essential for hip replacement surgeons, as it is a way to avoid postoperative complications, such as prosthesis dislocations, range of movement limitations, or prolonged pain after surgery. The crucial element of performing the surgery with a result satisfactory for the patient is preoperative planning. The algorithms mentioned above may be useful tools for a surgeon in the decision making process. Hopefully, a complete understanding of spinopelvic relations will increase satisfaction rates among patients after total hip replacement and contribute to further improvements in operation technique.

## Figures and Tables

**Figure 1 jcm-10-03528-f001:**
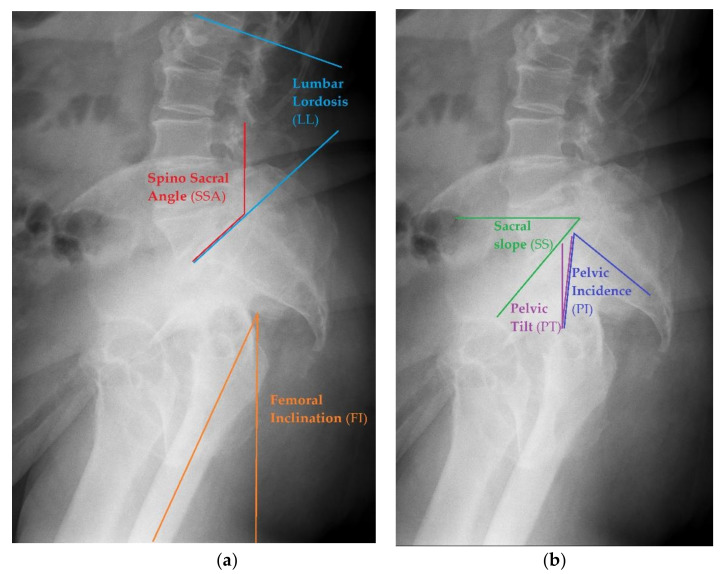
Examples of different pelvic measurements performed on lateral X-rays of the pelvis with lumbar spine view. (**a**) Spino Sacral Angle, Lumbar Lordosis and femoral Inclination presented on lateral X-ray. (**b**) Sacral Slope, Pelvic Incidence and Pelvic Tilt presented on lateral X-ray.

**Table 1 jcm-10-03528-t001:** Spinopelvic mobility types.

Spinopelvic Motion	Pelvic Tilt Change	Hip Bend Change	Lumbar Lordosis	Pelvic Femoral Angle
Normal	20°–35°	55°–70°	20°	45°
Hypermobile	>35°	<55°	>20°	<45°
Stiff	<20°	>70°	<20°	>45°

**Table 2 jcm-10-03528-t002:** Clinical consequences and problems in relation to pelvic position.

Types of Pelvic Position	Clinical Consequences	Potential Problems
In case of android type of pelvis (low PI, low SS, lower rotational movements) there is a tendency to retroversion, which leads to sacroiliac joints stiffness.	Low spinopelvic mobility is compensated by movement in hip joints.	Lower acetabular coverage -> risk of posterior dislocation during sitting
In case of a gynecoid type of pelvis (high PI, high SS, anteversion, higher rotational movements, no osteoarthritis in lumbar spine).	Spinopelvic mobility is restricted by hip joint movements. Lower extension in hip joints.	Higher possibilities of adaptive changes and higher acetabular coverage -> risk of anterior dislocation during standing
In case of lumbar spine stabilization with pelvis, with tendency to anteversion.	Higher acetabular anteversion.	Risk of anterior dislocation during standing
In case of lumbar spine stabilization with pelvis, with tendency to retroversion (flat back).	Lower acetabular anteversion.	Risk of posterior dislocation during sitting

PI, pelvic incidence; SS, sacral slope.

**Table 3 jcm-10-03528-t003:** Recommendation during THR (total hip replacement) in connection to spinopelvic alignment.

Spinopelvic Alignment	Preoperative Recomandation
Pelvic retroversion	Higher acetabular anteversion and inclination during THR—within limits of Lewinnek’s safe zone
Pelvic anteversion	Lower acetabular anteversion and inclination during THR—within limits of Lewinnek’s safe zone
Lumbar spine stabilization with pelvic with tendency to anteversion	Higher acetabular anteversion during THR—up to 30°
Lumbar spine stabilization with pelvic with tendency to retroversion	Lower acetabular anteversion during THR—up to 5°
Patients with hypermobile spinopelvic junction	Higher acetabular anteversion during THR—up to 25°
Patients with stiff spinopelvic junction	Higher acetabular inclination during THR—within limits of Lewinnek’s safe zone

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
