# Peer review of "Spinopelvic Alignment and Its Use in Total Hip Replacement Preoperative Planning—Decision Making Guide and Literature Review"

_jcm, 2021, doi:10.3390/jcm10163528_

Round 1

Reviewer 1 Report

JCM-1315530

Title: Spinopelvic alignment and its use in total hip replacement preoperative planning - decision making guide

Many thanks to the authors for having presented their review paper about guideline for spinopelvic alignment evaluation in THA. The topic is really interesting and actual. When considering manuscripts for publication, one of the key requirements for me as reviewer is that the paper must make a clear theoretical contribution to the literature. I think it could be very useful as a base for all surgeons who want to better understand this argument. The Author gives a clear and brief explanation of spinopelvic measurements and their relevance.

The abstract is adequate; materials and method described, but with lacking data. In my opinion the manuscript should be revaluated and can be accepted for publication with some minor revisions and one major revision.

Minor revisions:

- definitely needs mother language revision

- definitely needs grammar revision

-  Abstract: "Total hip replacement (THA)", should be either "total hip arthroplasty" or "THR". Also, in the rest of the manuscript, you use both THR or THA. Please, choose one for all, and then always use it.

- page 3. Figure (without number...) is way too misleading; also, in the description of P.I. (Pelvic incidence) you missed one of the two lines that defines PI: one is the line you described, the other is the perpendicular to the sacral superior endplate.

- line 175-176: "cup inclination changes 0.3 degree with 1 degree of PT". In which direction the change occurs? Is it a direct or inverse relation?

Major revision:

- The paper is substantially a review of the literature but there isn't any statement on how you performed the choice of the mentioned articles as well as a PRISMA should be reported. Moreover, Authors reported nor any statement on their work or any results of their studies. Of course, we trust you, but in my opinion it is not sufficient write a reference list only and citations in the text to define a manuscript a “review article”. 

Author Response

Thank you very much for helpful adivises and suggestions.

  1. Language revision was made
  2. THA was changed to THR as it is in whole manuscript
  3. Figures with radiological measurements were changed into more clear 
  4. We gave more accurate information about relation
  5. Manuscript was changed and new paragraph was added with detailed description on literature choice. As the article was not mentioned to be systemetic review, we did not provide PRISMA, if it is necessary we would make some more changes and provide it. Short section with our observational finding was added

Reviewer 2 Report

I commend the authors for a very popular topic in the total hip arthroplasty literature however I do have some suggestions and edits that should be considered

Section 3

*Line 96 – Although spine dynamics are a risk factor/source of dislocations that occur after primary THA with cup placement in the safe zone, it is not the only factor here

Section 6

*In order to be a more impactful decision making guide, when you mention suggestions like “higher acetabular anteversion and inclination during THA” you should give a general recommended range, if available.  This helps define what “higher” and “lower” are for the physician/surgeon.  For example, does “higher” imply OUTSIDE the safe zone range? Upper limit of normal? Etc etc

Section 7 / Discussion:

*it is important in this section to comment on the temporal relationship which is noted in the literature.  Especially the LONGER the patient is status post spinal fusion, the levels involved, and plus or minus fusion to the pelvis are big contributors to the risk of dislocation

*There are multiple typos in this section please edit and fix.  Please also improve English grammar content.  For example line 190 – in the setting of minimal inclination change from sitting to standing a surgeon can consider dual mobility to reduce dislocation risk

*Citation is needed for DMC

*May be worth adding a section here regarding navigation and/or robotic-assisted?   Preoperative planninga and software continues to get more and more sophisticated with virtual modeling and calculating the aforementioned parameters you have discussed in this paper

Author Response

Thank you very much for helpful advises and suggestions.

  1. We put greater emphasis on another risk factors
  2. The recomandations given are now more accurate
  3. More accurate information about spine fusion and its relation to THR was discussed
  4. English grammar was revised
  5. Citation for DMC added
  6. Our team considered robotic assisted navigation and its use in preoperative planning, but since the topic is quite complicated, we decided that it deserve separate paper to discuss every aspect